# The effect of cue length and position on noticing and learning of determiner agreement pairings: Evidence from a cue-balanced artificial vocabulary learning task

**Daniel R. Walter** [ID]**, Galya Fischer, Janelle Cai** *

Humanities Division, Emory University, Oxford College, Oxford, Georgia, United States of America

* daniel.walter@emory.edu

## Abstract

The importance of cues in language learning has long been established and it is clear that cues are an essential part of both first language (L1) and second/additional language (L2/A) acquisition. The effects of cue reliability and frequency, along with the competition between cues have been shown to significantly impact learners' pace of acquisition of these language-specific patterns. However, natural languages do not allow for a clear picture of how the forms of cues themselves affect their perception, uptake, and generalizability. In this study, we developed an artificial vocabulary consisting of determiners and nouns. Within these nouns, completely reliable cues were developed and equally distributed as long and short cues over three possible positions: beginning, middle, or end. Through a word-pair learning study, we show that length and position of cues variably affects agreement accuracy, and that noticing of cues during training is less important for known words, and more important for novel ones when deciding on inter-word gender-like agreement.

## Introduction

Linguistic cues (hereafter cues) are aspects of a linguistic code that point to meanings or relationships between words, and aid in comprehension and prediction of language use and learning. From the conception of the Competition Model [1, 2] and its evolution in the decades since [3–8], it is clear that cues are an essential part of both first language (L1) and second/additional language (L2/A) acquisition and use.

Studies on cue-based learning have often incorporated observational, corpus, and experimental data from real human languages, as well as artificial languages that have endeavored to discern the impact that these cues have on language learning in more controlled linguistic systems. These studies have shed light on the ways that cues affect the acquisition of verbal morphology [9, 10], morphosyntactic agreement [11–13], syntactic structure [14–16], and even long-distance dependencies [17, 18].

One issue with cue studies on natural languages has been that they do not easily allow for a straight-forward picture of how the forms of cues themselves affect their perception, uptake,

**Funding:** The author(s) received no specific funding for this work.

**Competing interests:** The authors have declared that no competing interests exist.

and generalizability. This is because, in natural languages, the cue-internal and cue-external factors are inextricable, thus extricating the individual influences of internal versus external only results in an opaque picture.

Cue-internal factors, such as its form (length/phonetics/graphemes) and position (beginning/middle/end of a word) can differ drastically within the same language. For example, cues indicating the same type of information (e.g., noun-class or number) can contrast in their form, with some being as short as a single phoneme and others spanning multiple syllables, or in their position, with some appearing at the beginning of a word and others at the end. Cue-external factors, such as frequency and reliability, can also vary wildly. Again, within the same language, one cue might be found thousands of times while another is found only in a handful of words. Or, a cue might be completely reliable at indicating a particular relationship or meaning, while another might only be reliable for a certain percentage or types of words.

Precisely because of the unbalanced nature of cues in natural languages, studies examining natural languages have been unable to fully isolate how the properties of cues affect the noticing and learning of agreement patterns. Knowing more about the impact of the structural elements of cues is essential to understanding the uptake of noun features and the eventual categorization of nouns into various subclasses.

One important dependency aided by cues is that of morphosyntactic agreement within noun phrases. In many languages, such as German, Spanish, Swahili, and Italian, among others, cues present on nouns provide information about grammatical gender (or noun-class), which is required to assign gender to constituent words and morphemes, like determiners and adjectives. While one can use rote-learning to memorize the gender of an individual noun, cues found within the noun often provide useful information about the appropriate gender-agreement marking, which allows speakers to generalize patterns across known words, as well as onto novel ones. However, the ease of recognition and use of these cues is linked to their variability, in both their cue-internal and cue-external properties. This variability often affects which cues are learned and noticed by L2 learners [19].

In order to isolate the effects of cue-internal features on the learning and noticing of agreement patterns from cue-external ones, this study utilizes a miniature artificial language consisting of determiners and nouns. The artificial language neutralizes the natural cue-external factors by holding distribution and reliability constant, while manipulating the cue-internal factors of length and position. In this way, this study provides new insight into the unique contributions of cue-internal factors to cue-learning and noticing.

The study investigated the following research questions:

1. Holding cue-external factors constant, do cue-internal factors of length and position, or a combination thereof, affect adult learners' ability to learn gender-like agreement between nouns and determiners?

2. Are participants who indicate noticing of cues to determiner-noun agreement after training more successful at assigning agreement than those who do not, and does the accuracy of the reported noticing matter?

## Literature review

The importance of cues for learning inter-word relationships has been well established. For example, studies have shown that learners can use distributional cues to segment words [20, 21] and categorize nouns [22, 23]. Saffran et al. [21] even point out that "adults are able to discover word units rapidly even in a system as impoverished as an unsegmented artificial

language," (p. 618), which provides explanatory power for humans' ability to acquire languages with disparate cue distributions, frequencies, and reliability. Additionally, findings that show the rapidity of learning for adults [24] indicate that these learning mechanisms are not only available to adults, but also quickly implemented (Hamrick, 2013 [unpublished], [22]). And while these distributional factors have been shown to be sufficient for word categorization "even without correlated phonetic or semantic cues" ([23] p. 371), others have shown that semantic [25] and phonological [20, 26, 27] properties of the input can and do serve as cues for learning about words, morphemes, and the relationships between them.

One common grammatical feature that requires multiple-word sequences for processing is noun-feature agreement. This type of agreement occurs between nouns and constituent words that appear within noun-phrases, such as determiners and adjectives. In these agreement structures, the noun and its constituents need to share some common information. It has been shown that these agreement structures are learned better when presented as multiword chunks, as compared to more segmented units [28–30].

While the current study focuses on noun-internal information shared across constituents, similar information sharing from hierarchical syntactic relations can also provide insight into how adults learn noun-phrase agreement structures. In two studies on syntax and case learning, Grey et al. [31, 32] used the artificial language "Japlish" adapted from Williams and Kuribara [33] to study whether adults could acquire Japanese-like case-marking from unsegmented speech. In this language, English words serve as the lexicon, but the sentences follow Japanese word order and have Japanese case markers. In the first study, learners were exposed to Japlish only orally. After the training phase, many of the participants were able to formulate rules about the syntactic relationship between words in Japlish [31]. This finding shows a sensitivity for hierarchical cues based on syntax, but no evidence for the use of collocational information from the case-marking particles as important cues to syntax. In the second study, the researchers controlled for individual differences, specifically the role of phonological working memory, learning styles, and personality. In this study, they also found that adults displayed cue-based learning without explicit instruction [32]. These studies are important for understanding the context of the current study, because, as Grey et al. [31] note, there has been "strikingly little research [. . .] on the extent to which effects of language learning under incidental exposure apply to more than just word order (e.g., morphological markers)," (p. 612). The difference in learning between experiment 1 and experiment 2 also highlights the variability in information processing, pattern recognition, and morphosyntactic learning that can arise based on individual and contextual factors.

As is the focus in this study, noun-internal features can also serve as a basis for noun-phrase agreement, and need to be known to form the appropriate agreement structures [13, 34]. While some of this noun-internal information may be structurally invisible (e.g., noun-class in Swahili based on semantic categorization), the phonological and morphological structure of nouns in languages with gender and noun-class systems often provide cues to gender. For L1 speakers, the repeated exposure to nouns with these cues via chunked agreement patterns results in a categorization of nouns into subclasses. Whether L1 speakers are aware of these morphosyntactic regularities or not, this knowledge aids in speakers' uptake, processing, and prediction of noun-phrase agreement. However, it is unclear to what extent L2 learners can make use of these morphosyntactic regularities, as their overall exposure to their L2 may not be enough to learn these patterns without explicit instruction.

In one study using ERPs, Morgan-Short and colleagues [35] investigated whether L2 learners could process agreement structures in similar ways to L1 speakers by testing their response to ill-formed determiner-noun agreement pairs of an artificial language. They found similar P-600 responses (an ERP response associated with processing ungrammaticality across

dependent words) for L2 learners at "both lower and higher L2 experience/proficiency levels, under both explicit (classroomlike) and implicit (immersionlike) training conditions," (p. 185). The importance of these findings is two-fold. First, similar neural processes were observed in L2 learners to those shown by L1 speakers while processing agreement patterns. And second, this neural process was elicited in an implicit training condition. These findings indicate that L2 learners are indeed capable of not just learning, but also processing noun-phrase agreement patterns.

Further evidence of L2 learners' use of cues to assign noun gender and apply it in noun-phrase agreement comes from a series of studies by Kempe, Brooks and colleagues [36–41]. In these studies, the researchers focused on the influence of phonological and morphological cues in Slavic languages on the acquisition of agreement structures. Results from these experiments repeatedly showed a positive relationship between the transparency, availability, and reliability of gender cue marking and correct identification and assignment of gender.

The series of studies by Brooks, Kempe, and colleagues [36–41] also provide some implicit direction regarding the role of cue-internal factors on learning. First, the diminutive marking in Russian is always at the end of the noun, and therefore, while comparisons to other positions cannot be made, the effectiveness of the diminutive marking as a cue to gender may be a result of its word-final position. As Brooks et al. [37] found, the consistent cues available through diminutive endings allowed learners to be more accurate in their generalization of case-marking and adjective-noun agreement compared to simple nouns without diminutive morphology. The authors argue that the form and the location of the diminutive made them salient, allowing learners to extract the highly reliable cue information and apply it to the relevant agreement pattern between the noun and determiner.

Another finding to highlight comes from Kempe and Brooks [39]. In this study, the researchers found that gender classification occurred faster for participants when they were exposed to words with diminutive marking, despite the diminutive nouns always being longer. The importance of these findings is multifaceted. First, it affirms that learners are relying on these multiword sequences to process agreement structures across words using cues. And second, the impact of these cues was not negatively affected by the amount of phonological complexity that they added to noun.

In sum, these studies provide a foundation for the importance of cue-internal factors to the ways in which both L1 and L2/A learners acquire noun-phrase agreement. However, they either lack sufficient control over cue-external factors or fail to manipulate cue-internal factors in a systematic way. The current study is motivated by this research gap and aims to paint a clearer picture of how cue-internal factors affect the learning of noun-phrase agreement.

## Methodology

On 3/22/2022, the Emory IRB reviewed this study (reference number STUDY00002093) by expedited process and deemed it eligible for expedited review under 45 CFR.46.110 and/or 21 CFR 56.110 because it poses minimal risk and fits expedited review category F [7] as set forth in the Federal Register. Consent was obtained online through a written form.

### Participants

Participants were recruited via an online advertisement through Facebook from March 25th, 2022 through April 23rd, 2022. In total, 54 participants (female = 35, male = 16, non-binary = 3) completed the entire study. All participants indicated that English was their first language (46 of 54 participants), or that they spoke English fluently as a second language (8 of 84 participants). Additional first languages included Chinese/Mandarin (4), Dutch (1), Hindi (3), Twi

(1), French (1), Gujarati (1), Portuguese (1), Slovak (1), Urdu (1), and Spanish (1). Note that some speakers indicated multiple L1s, so the sum of languages here does not equal the number of participants. Of the 54 participants, 11 indicated knowing a third language. These additional languages included Chinese/Mandarin (4), French (14), Spanish (27), Czech (1), Russian (1), Japanese (2), German (6), Korean (2), Latin (3), Hebrew (3), Italian (3), Punjabi (1), Euskera (1), Tamil (1), Hindi (4), and Urdu (3). Participants under 18 years of age were immediately rejected through Gorilla Experiment Builder's [42] branching logic. Participants provided consent through the Gorilla platform on the first page of the study and this consent was recorded and required to continue on with the study. The average age of the participants was 21.43 years (SD = 6.28 years). This distribution was right tailed and had a wide standard deviation of 6 years.

## Materials

The experiment was presented online through Gorilla Experiment Builder [42]. The materials for the study consisted of a background questionnaire, a determiner-word agreement learning phase, a posttest for known words, a posttest for novel words, and a debriefing questionnaire.

The items for this study were created to balance long and short cues across three positions: beginning, middle and end. Each of these cues was assigned one of three possible determiners which were always a reliable match between determiner and cue. The resulting six cues to agreement, the matching determiner, and an example word are provided in Table 1 below along with their length and position within the noun. Length was operationalized as either long or short, where short cues contained only one letter/phoneme, while all long cues contained as letters a vowel and at least two consonants, and phonetically at least one vowel and two distinct consonants, giving them their own syllable structure within the noun.

Each agreement cue was incorporated into five artificial nouns. Each of these nouns was workshopped to ensure that no competing cues could be found within the words and that they all followed legal phonetic patterns in English. The same was done for a set of novel (untrained) nouns that would be used in the second posttest. This resulted in a total of 60 determiner-noun pairings, with 20 words total being assigned to the *mo*, *lu*, and *zi* determiners (see S1 Table for complete list of stimuli).

The first posttest consisted of a matching and selection task with the words from the training phase. Participants were given a word-bank for all the trained nouns, without their determiners, in the upper half of the screen. In the middle of the screen, participants saw the English translation. And at the bottom of the screen, participants were asked to fill in two pieces of information. On the lower middle-right-hand-side, participants were asked to write in the artificial noun for the English noun presented in the middle of the screen and to use the word-bank above. On the lower left-hand-side, participants were asked to select the appropriate artificial determiner that matched the artificial word they had written. Items in this task

**Table 1. Stimuli cue and determiner description.**

| Determiner | Cue | Length | Position | Example |
|---|---|---|---|---|
| mo | grup- | long | beginning | "grupswek" |
| zi | a- | short | beginning | "arotep" |
| lu | -sul- | long | middle | "resulm" |
| mo | -y- | short | middle | "kemypo" |
| zi | -shre | long | end | "megleshre" |
| lu | -f | short | end | "vref" |

were presented in random order. The experiment platform tallied the correct responses for English-artificial noun matching and presented the number correct after completing all 30 known items. The second posttest consisted of a determiner selection task only, consisting of 30 novel words. The artificial novel nouns were presented in random order, one at a time, in the center of the screen, with a selection box for the artificial determiner located to the noun's immediate left. The final task asked three questions intended to assess participants' noticing of patterns in the data. The first question asked whether and what patterns participants noticed in the data. The second question asked whether anything they noticed helped them to match the determiners and nouns. And the final question asked whether they thought their experiences with any of their languages (L1s or L2s) played a role in their ability to see these patterns.

## Procedure

Upon clicking the study link, participants were directed to a study overview, including the approved IRB language, and asked to check a box if they agreed to participate in the study. On the second page, participants provided their date of birth, gender, and information about which languages they knew. After clicking the "next" button, the study branched to check that all participants were over the age of 18. If they were under 18, they were directed to a page explaining that they could not participate in the study. If they were over 18, they were directed to a page that overviewed the order of the training and testing materials.

After hitting the "next" button, participants advanced to a page explaining the training phase. They were instructed that would be presented with two words that they had to memorize, and that they would be tested on both words in the posttest. The training items were only presented in writing. It was emphasized that they needed to get both words correct in order to receive points in the posttest. However, they were not told, at any point, that the artificial nouns contained cues that could help them with agreement between the determiner and noun. After clicking "next" again, participants were given three examples of how the training phase would proceed, including the "+" fixation points and English/artificial word pairs, along with the same timing used to move through items that they would experience in the training phase. Fig 1 below provides an example of the training procedure and word reveal.

After these practice items, participants were asked to click the "next" button when they were ready to begin the training phase.

The training phase consisted of the fixation and word-reveal procedures. Participants were presented with a screen that contained two "+" fixation marks, one centered in the top half of the screen, and the other in the bottom half of the screen. After 0.5 seconds, the "+" in the upper half of the screen was replaced by the artificial determiner-noun pairing. After another 0.5 second, the "+" in the lower half of the screen was replaced by the English translation ("the [item]"). Both the artificial determiner-noun pair and the English translation remained on the screen for 3.0 additional seconds before moving on to the next item. The next item began with the initial screen containing the two "+" fixation marks and followed this pattern until all 30 items had been completed. The task was designed so that items were presented in a random order to prevent ordering and recency effects in the posttest.

These training items were repeated in random order 5 times. After all 30 items had been presented for a single trial, the program presented the number of training trials remaining. Upon completion of the fifth training trial, participants were asked to click "next" when they were ready to proceed to the testing phase.

Before beginning the first posttest, participants were provided instructions about how to complete the test items. All items from the word-bank remained for every item and did not disappear once they were used. Test items were presented one-by-one in a random order.

lu tesulni          lu tesulni

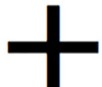          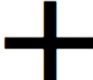          the apple

**Fig 1. Training and testing word reveal procedure.**

After completing all 30 items, the program reported the number of nouns they had correctly matched, regardless of whether the determiner they had selected matched the artificial noun they had written.

After viewing their results and clicking "next", participants moved on to the second post-test. Again, participants were presented with the test items in random order. As all the nouns were novel, there was no matching task or word bank. Participants were only asked to select the determiner they thought matched the novel noun. Upon completion of the 30 items, participants were presented with the posttest questionnaire.

For the posttest questionnaire, participants were asked to complete, in as much detail as they liked, the questions described above in the materials section. Then, they were provided a debrief about the study that included information about the connections between the artificial noun cues and the determiners, and about the reason they had been deceived about the true nature of the study. After clicking "next", participants were directed to the final screen that collected their payment information and then logged out of the experiment. A time-limit for completion of the study was set so that participants who took more than two hours to complete the experiment were automatically timed out. The average time for this study to be completed was approximately 30 minutes.

## Analysis

Before analyzing the data, the individual responses to the posttest questions related to cue noticing were assessed. If a participant indicated that they did not notice any patterns in the data, they were assigned to the group No Noticing. If they indicated that they did notice patterns, but the patterns they indicated were incorrect, i.e., not cues used in the experiment, they were assigned to the Incorrect Noticing group. And if they indicated that they did notice patterns and the participant was able to name at least one cue used in the data and say which determiner it was paired with, they were assigned to the Correct Noticing group. This categorization was used as a factor in the statistical analysis.

The primary analysis for this experiment were two mixed-effects binary logistic regressions (MEBLRs). The first MEBLR, given in Expression 1 below, was used to model the effect of the fixed-effect variables of cue position (ρ), cue length (λ), noticing (ν), and the word matching accuracy for known words (κ) on known word agreement accuracy (Z). Participant ID (ι) was added as a random effect to better allocate the variance due to the repeated measure of word agreement. Cue position (ρ) was coded as a 3-level factor corresponding to beginning, middle, and end. Similarly, cue length (λ) was coded as a 2-level factor corresponding to short and long. Noticing (ν) was coded as a 3-level factor, including levels of no noticing, inaccurate noticing, and correct noticing. Finally, the noun matching accuracy for known words (κ) was a continuous variable using the raw scores from the matching test.

*Mixed-effects logistic binary regression formula for known words*

$$Z(\text{known}) = \log\left(\frac{p}{1} - p\right) = B0 + \rho P + \lambda \Lambda + \nu N + \kappa K + I\iota + e \tag{1}$$

The second MEBLR, given in Eq 2 below, was used to model the effect of the variables of cue position (ρ), cue length (λ), and noticing (ν) on novel word agreement accuracy (Z). Participant ID (ι) was again added as a random effect. In this model, since none of the novel test items had a score for word matching accuracy, this variable was dropped. Variables were coded the same as in the first MEBLR.

*Mixed-effects logistic binary regression formula for novel words*

$$Z(\text{novel}) = \log\left(\frac{p}{1} - p\right) = B0 + \rho P + \lambda \Lambda + \nu N + I\iota + e \tag{2}$$

Multiple models for both MEBLRs were compared using AIC scores to find the best fit by testing for main effects of cue length and position independently of one another, for mixed effects of cue length and position, and with and without the noticing variable.

All statistics were conducted in *R* [43] with additional packages [44–48]. All data and code required to reproduce all analyses are available at https://osf.io/45hvk/?view_only= 9bc41b76a59a4dd7837000f428e50e5a, found in the folder named Revised Code for PLOS One. Use R-code CueDataR3.5.24.R and attached Excel files.

## Results

The descriptive statistics show a wide variance in participants' ability to match the artificial nouns with their English translations ($N = 54$, $M = 12.11$, $SD = 9.36$, 95% CI [9.56, 14.67]). Participants' ability to correctly match the artificial noun to the artificial determiner was above chance at 39.2%, compared with expected 33.3% chance of randomly guessing the determiner. In addition, a Chi-squared test found no statistical differences for agreement accuracy between the three determiners *lu*, *mo*, and *zi* ($\chi2 = 0.365$, $df = 2$, $p = 0.833$). Agreement accuracy was also above chance, where guessing at determiners would lead to an expected 10 out of 30 (see confidence intervals), and on average higher for known (trained) words ($M = 13.13$, $SD = 3.39$, 95% CI [12.21,14.05]) than for novel (untrained) words ($M = 11.29$, $SD = 4.21$, 95% CI [10.15,12.44]). The difference in agreement accuracy between known and novel words was significant ($t = 2.502$, $df = 106$, $SE = .0736$, $p = .014$). Finally, in order to outline differences in the Notice variable, we tallied differences between participants answers to the questions on the posttest debrief on their noticing ability. From the total group of participants ($N = 54$), 27 indicated no noticing whatsoever, and 27 indicated some level of noticing. Of the latter group of noticers, 12 were incorrect noticers and 15 were correct noticers.

Moving to the regression analyses, the first MEBLR, provided below in Tables 2 and 3, tested models for main effects of all variables and of models with an interaction between the

**Table 2. Model 1: Known word agreement accuracy (Main fixed-effects only).**

| Model | 1a | 1b | 1c | 1d | 1e | 1f | 1g | 1h | 1i | 1j | 1k |
|---|---|---|---|---|---|---|---|---|---|---|---|
| AIC | 2199 | 2154 | 2199 | 2156 | 2178 | 2141 | 2178 | 2142 | 2180 | 2142 | 2143 |
| (Intercept) | -0.22* (0.10) | -0.53*** (0.10) | -0.32** (0.12) | -0.60*** (0.12) | -0.09 (0.09) | -0.43*** (0.10) | -0.19 (0.11) | -0.49*** (0.11) | 0.01 (0.11) | -0.32** (0.12) | -0.39** (0.13) |
| Position (middle) | -0.09 (0.12) | -0.12 (0.12) | -0.09 (0.12) | -0.12 (0.12) | --------- | --------- | ---------- | ---------- | -0.09 (0.13) | -0.11 (0.13) | -0.12 (013) |
| Position (end) | -0.19 (0.13) | -0.20 (0.13) | -0.19 (0.13) | -0.20 (0.13) | --------- | --------- | ---------- | ---------- | -0.19 (0.13) | -0.20 (0.13) | -0.20 (0.13) |
| Length (long) | ---------- | ---------- | --------- | ---------- | -0.47*** (0.10) | -0.39*** (0.10) | -0.47*** (0.10) | -0.39*** (0.10) | -0.47*** (0.10) | -0.39*** (0.10) | -0.39*** (0.10) |
| Noun Matching Accuracy | ---------- | 0.78*** (0.11) | --------- | 0.76*** (0.11) | ---------- | 0.73*** (0.11) | ---------- | 0.71*** (0.11) | --------- | 0.74*** (0.11) | 0.71*** (0.11) |
| Notice (inaccurate) | ---------- | ---------- | 0.02 (0.16) | 0.05 (0.15) | ---------- | ---------- | 0.02 (0.18) | 0.05 (0.15) | --------- | --------- | 0.05 (0.15) |
| Notice (accurate) | ---------- | ---------- | 0.33* (0.16) | 0.22 (0.14) | ---------- | ---------- | 0.33* (0.16) | 0.23 (0.14) | --------- | --------- | 0.23 (0.14) |

Standard Error indicated in (), significance codes:

\* < .05,

\*\* < .01,

\*\*\* < .001, best model fit indicated by

**Table 3. Model 1: Known word agreement accuracy (Fixed-effects with position and length interaction).**

| Model | 1l | 1m | 1n | 1o |
|---|---|---|---|---|
| AIC | 2183 | 2145 | 2183 | 2147 |
| (Intercept) (Position (beginning):Length (short)) | 0.08 (0.13) | -0.27 (0.14) | -0.02 (0.15) | -0.34* (0.15) |
| Position (middle):Length (short) | -0.23 (0.18) | -0.21 (0.18) | -0.23 (0.18) | -0.21 (0.18) |
| Position (end):Length (short) | -0.26 (0.18) | -0.25 (0.18) | -0.26 (0.18) | -0.25 (0.18) |
| Position (short):Length (long) | -0.61*** (0.18) | -0.49** (0.18) | -0.61*** (0.18) | -0.49** (0.18) |
| Position (middle):Length (long) | 0.29 (0.25) | 0.19 (0.25) | 0.28 (0.25) | 0.19 (0.25) |
| Position (end):Length (long) | 0.15 (0.25) | 0.11 (0.25) | 0.15 (0.25) | 0.11 (0.25) |
| Noun Matching Accuracy | ---------- | 0.73*** (0.11) | ---------- | 0.71*** (0.11) |
| Notice (inaccurate) | ---------- | ---------- | 0.02 (0.17) | 0.05 (0.15) |
| Notice (accurate) | ---------- | ---------- | 0.33* (0.17) | 0.23 (0.14) |

Standard Error indicated in (), significance codes:

\* < .05,

\*\* < .01,

\*\*\* < .001

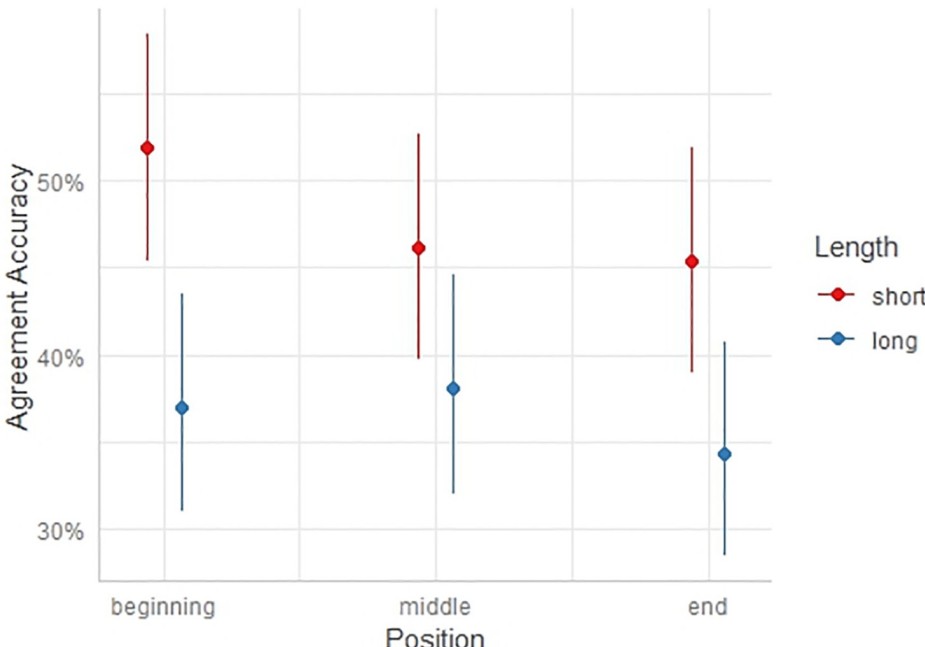

**Fig 2. Predicted probabilities of known word accuracy as an interaction between cue position and length.**

Position and Length variables. Additionally, Fig 2 depicts the predicted probabilities of known word agreement accuracy as an interaction between cue position and length.

Fig 2 reflects the lack of interaction between cue position and length for agreement accuracy between known words and determiners, supporting the adoption of a main-effects-only model.

The best fit model as indicated by AIC scores was Model 1f, with Length as a main effect without Position. Other close models included Models 1h, 1j, and 1k. All these models indicate high significance for the Length and Noun Matching Accuracy variables. In these models that included Position (Models 1j and 1k), Position was not significant. Of these models, Model 1f is preferred because it has fewer variables and all of its variables are significant. In Model 1f, longer cues are associated with a 32.3% decrease ($1-e^{-0.39} = 0.323$) in the odds of assigning the correct determiner to the known noun. Additionally, every increase of 1 point in the translation task score was associated with an increase of 107.5% ($e^{0.73}-1 = 1.075$) in the odds of assigning the correct determiner to the know noun.

The analysis of the MEBLR for novel words took the same approach. Results displayed in Tables 4 and 5 below tested models for main effects of all variables and of models with an interaction between the Position and Length variables. Fig 3 depicts the predicted probabilities of known word agreement accuracy as an interaction between cue position and length.

Fig 3 reflects the interaction between cue position and length, where longer cues provide significant improvement in agreement accuracy between novel words and determiners when the cue is in the middle of a word. However, this difference in length does not appear to affect cues at the beginning or end of words.

There are two models with AICs that fall below 2080, namely Models 2b and 2g. The difference between these two models is slight, although the AIC score for Model 2g is lower, which is one indicator that it is a better fit. Starting with Model 2b, we see a significant effect for both

**Table 4. Model 2: Novel word agreement accuracy (Main fixed-effects).**

| Model | 2a | 2b | 2c | 2d | 2e |
|---|---|---|---|---|---|
| AIC | 2083 | 2078 | 2094 | 2089 | 2080 |
| (Intercept) | -0.26*<br>(0.11) | -0.49***<br>(0.13) | -0.53***<br>(0.10) | -0.76***<br>(0.12) | -0.53***<br>(0.14) |
| Position (middle) | -0.26*<br>(0.13) | -0.26*<br>(0.13) | - - - - - - - - - - | - - - - - - - - - - | -0.26*<br>(0.13) |
| Position (end) | -0.46***<br>(0.13) | -0.46***<br>(0.13) | - - - - - - - - - - | - - - - - - - - - - | -0.46***<br>(0.13) |
| Length (long) | - - - - - - - - - - | - - - - - - - - - - | 0.07<br>(0.11) | 0.07<br>(0.11) | 0.07<br>(0.11) |
| Notice (inaccurate) | - - - - - - - - - - | 0.42*<br>(0.19) | - - - - - - - - - - | 0.41*<br>(0.19) | 0.42*<br>(0.19) |
| Notice (accurate) | - - - - - - - - - - | 0.51**<br>(0.18) | - - - - - - - - - - | 0.50**<br>(0.18) | 0.51**<br>(0.18) |

Standard Error indicated in (), significance codes:

\* < .05,

\*\* < .01,

\*\*\* < .001

Position (middle) and Position (end), as contrasts with the intercept Position (beginning). We also see a significant difference between the noticing group levels of Notice (inaccurate) and Notice (accurate), and the intercept level of No Noticing. Assessing the effects of Position, cues in a middle position were associated with a 38.7% decrease (1-e^-0.49 = 0.387) in the odds of assigning the correct determiner to the novel noun, and cues at the end of words were

**Table 5. Model 2: Novel word agreement accuracy (Fixed-effects with position and length interaction).**

| Model | 2f | 2g |
|---|---|---|
| AIC | 2081 | 2076 |
| (Intercept)<br>(Position (beginning):Length (short)) | -0.18<br>(0.14) | -0.42**<br>(0.16) |
| Position (middle):Length (short) | -0.60**<br>(0.18) | -0.60**<br>(0.18) |
| Position (end):Length (short) | -0.47*<br>(0.18) | -0.47*<br>(0.18) |
| Position (beginning):Length (long) | -0.15<br>(0.18) | -0.15<br>(0.18) |
| Position (middle):Length (long) | 0.66*<br>(0.26) | 0.66*<br>(0.26) |
| Position (end):Length (long) | 0.02<br>(0.26) | 0.01<br>(0.26) |
| Notice (inaccurate) | - - - - - - - - - - | 0.42*<br>(0.19) |
| Notice (accurate) | - - - - - - - - - - | 0.51**<br>(0.18) |

Standard Error indicated in (), significance codes:

\* < .05,

\*\* < .01,

\*\*\* < .001, best model fit indicated by

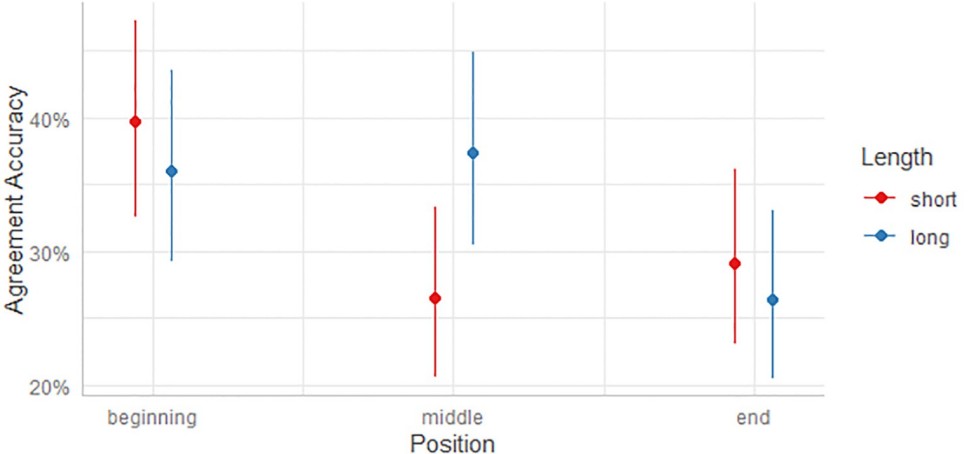

**Fig 3. Predicted probabilities of novel word accuracy as an interaction between cue position and length.**

associated with a 22.9% decrease (1- e^-0.26 = 0.229). In addition, increases in noticing were associated with an increase in the odds of assigning the correct determiner to the novel noun. Incorrect noticing was associated with an increase of 52.2% (e^0.42–1 = 0.522) in the odds of selecting the correct determiner, and correct noticing was associated with an even larger increase of 68.2% (e^0.52–1 = 0.682).

Turning to Model 2g, we see significance for the following interactions: Position (middle): Length (short), Position (end): Length (short), Position (middle): Length (long). There was a relative decrease in odds for assigning the correct determiner to the novel noun for the interactions Position (middle): Length (short) (45.1% decrease (1-e^-0.60 = 0.451)) and Position (end): Length (short) (37.5% decrease (1-e^-047. = 0.375)), while there was a relative increase in the odds as compared to the intercept Position (beginning): Length (short) for the interaction Position (middle): Length (long) (93.5% increase (e^0.66–1 = 0.935)). As in Model 2b, Model 2g also shows significance for the two Noticing levels against the intercept level of No Noticing, with similar degrees of effect. Considering the AIC scores, the interaction between Position and Length, the number of significant variables, and the consistency of signs and significance for main effects between the two models, Model 2g with the interaction between Position and Length is preferred and will be used for the discussion of the results.

## Discussion

This study aimed to uncover links between cue-internal factors, specifically their length and position within a noun, and individuals' ability to learn gender-like agreement patterns between nouns and determiners. The first research question asked whether cue length or position, or a combination thereof affected learners' ability to correctly assign gender-like agreement onto determiners. This study provides different answers to this question when comparing the results of the known-word and novel-word stimuli. The MEBLR for known words indicated no interaction between cue length and position. Words with short cues were more likely to be assigned the correct determiner as compared with those with longer cues, and position did not play a significant role in participants' ability to assign the correct determiner to known nouns. However, the ability for this model to answer questions about position and length of cues on noun-agreement learning is complicated by prior word knowledge, which was the greatest predictor of agreement accuracy. It is possible that trained words with

short cues were easier to remember because they were shorter and therefore less phonologically complex than trained words with long cues. The higher accuracy score for agreement between determiners and nouns with short cues thus entangles prior word knowledge with cue length.

The MEBLR for novel words provides a different picture of the effects of cue length and position without the mediating variable of prior word knowledge. Here, position, in addition to length, played a significant role; there were better odds when the cue was at the beginning of the word, then in the middle, and worst odds when the cue was at the end of the word. This indicates that the proximity of cues to their interrelated words plays a role in their uptake and utility. The immediate position between determiners and cues in the beginning position showed the best generalization to novel words, while the farthest distance, that between determiners and cues in the end position, showed the lowest generalization.

These second important finding from the novel-word data is the interaction between length and position. In the best fitting model, there was a positive interaction effect for cues that were both long and in the middle of the novel nouns on participants' ability to select the agreeing determiner. This positive effect on length for cues in the middle of nouns makes sense, as the increased phonological/orthographic length of the cue may aid in saliency and therefor processing or noticing the cue. Of note here is that none of the languages spoken by participants utilize noun-internal positions for gender-like cues, and these languages typically store morphosyntactic information at the beginning or end of words. Therefore, participants' entrenched habits of looking for morphosyntactic information at the beginning or end of nouns may have made additional saliency through long cues unnecessary, while the uncommon middle position of cues in this data set became more salient through their length.

The findings from the novel-word data stand in contrast to some prior studies [49, 50], where there was a preference for suffixation over prefixation in the categorization of nouns. However, in the case of this study, the cues were not used to discern between nouns, they were required to be used across a longer distance, that between the determiner and the noun. This may be why the cues in the beginning position resulted in a higher chance of agreement accuracy. Therefore, the links between the determiners *zi* and the nouns that started with *a-* (and similarly for *mo* and *g(rup)-*) were being learned because of their proximity, despite the preference in English for using suffixes over prefixes to categorize nouns.

Another reason for the difference between this study and the Martin and Culbertson [49] and Ramscar [50] studies may be that these cues are purely phonological in nature, and therefore not really affixes to the artificial words. In that sense, the lack of morphological significance may play a role in driving the different results found in this study. The differences in accuracy by syllable length may also be due to the differences in phonological properties and syllabic structure, which has been shown to affect multiword sequence learning [27].

For the second research question, we were interested in measuring differences in agreement accuracy between participants who indicated noticing of cues and patterns, and those who did not. Using the posttest debriefing form, we parsed participants into three groups, namely those who reported no noticing, incorrect noticers, and correct noticers. The results from the known-word data do not show an association between either group of noticers and increased or decreased odds of correctly assigning determiners to known nouns. This is likely due to the strong effect we see of word knowledge as measured by the matching task. Contrastingly, the results from the novel-noun data show a strong association between noticing and the odds of assigning the correct determiner to a novel noun, with the odds for correct noticers higher than those for incorrect noticers. This finding is likely indicative of the importance of noticing as being relative. For known words, reliance on rote-memorized determiner-noun pairs enabled all learners, irrespective of their noticing of cues, to be able to supply accurate

agreement. For novel nouns, if the participant did not have any awareness of the cues, they would be guessing at random, while the participants who indicated noticing, and even more so the participants who correctly noticed the connections between certain cues and determiners would have a significant advantage. While it is not possible to consider dual routes to agreement (rote vs. generalized patterns), this finding is important for understanding new word learning. The data suggest that participants who indicated noticing cues were more capable of applying that knowledge to novel items than participants who did not indicate noticing cues. These findings are in line with other studies that have shown significant advantages for learners who indicate noticing cues or patterns over those who do not (Hamrick, 2013 [unpublished], [38, 51]).

The pedagogical implication of these findings is also important to discuss here. Increasing evidence has emphasized the role for explicit instruction for complex grammatical features, but the amount of time and resources available within the classroom are limited. Using the findings from this study, we can gain a better understanding about which features learners are likely to be able to learn and/or notice on their own. Educators could spend more time on patterns which are not likely to be learned implicitly or noticed through awareness raising tasks or other form-focused instruction. For agreement features specifically, one might be able to use this information to predict with cues, based on position and length, might be more noticeable to learners.

## Limitations

There are a few limitations of this study. First, the determiner-cue pairs were not randomized, and because of the structure of the data, it is not possible to tell if certain determiner-noun pairs or determiners themselves were simply easier to remember, even though analyses indicated no differences between accuracy of the three determiners overall. Further studies focused on structural similarities between determiner-noun pairs would provide insight in this area. A second limitation of this study is the generalizability of these findings to speakers of other languages beyond English speakers. Most participants indicated that their first language was English, and all other participants indicated knowing English as an L2. Therefore, the finding that cues in at the beginning of nouns were both noticed and learned better than cues in the middle or at the end of nouns may stem from noticing behaviors in English. English speakers are also not generally used to looking for gender-like cues between determiners and nouns, as this does not provide any processing support in English. In English, agreement patterns do exist between nouns and determiners, but to a much lesser extent, e.g., on a phonological basis for deciding between *a/an*, number agreement like in *this/these*, and count versus non-count nouns like *much/many*. In this regard, L1 English speakers do not use determiner-noun agreement to categorize nouns into noun-classes as this artificial language does.

## Future directions

Future studies will attempt to overcome this L1/L2 limitation and tease out the influence of L1/L2 noticing behaviors by comparing various language groups. For example, we would like to investigate Turkish speakers, who are conditioned to pay attention to infixes, which occur frequently in Turkish. A comparison of English speakers to German, Spanish, or Italian speakers would also help to understand how speakers of L1s who are accustomed to use word endings to support agreement cues differs from speakers who do not continually process determiner-noun agreement structures. A methodological innovation to the study, specifically the inclusion of eye-tracking, would allow us to capture differences in where and how long L1 groups differ by looking behaviors. This future work would provide an interesting point of

comparison to Brooks and Kempe [38], who found that while the learning of case information was best predicted by levels of learner awareness, the "learning of gender agreement was best predicted by the learners' familiarity with other languages with similar gender systems." This finding regarding the importance of language similarity and transfer was also reflected in Franck et al. [13].

In addition to exploring L1/L2 differences, understanding the effect of the position of the determiner would allow us to disentangle the impact of the cue proximity to determiner in the visual field. For example, if we were to use a postposition for determiners, we would be able to see whether this increases noticing of cues at the end of words. Additionally, we would want to compare speakers who are accustomed to looking for determiner-like words in post-noun positions, such as Japanese, where case marking appears after the noun.

## Conclusion

Overall, this study showed that the learning of determiner-noun agreement based on noun-internal cues by adult learners, when held consistent and reliable, are affected by the length and position of the cue and prior word knowledge, in addition to noticing. While earlier studies have shown that the role of the linguistic structure itself is significant [52, 53] in addition to recent findings on the role individual differences on the learning of agreement structures through implicit or incidental means [32, 34, 38], it was difficult to assess how the form and position of cues as isolated variables could affect learning.

This study has shown that isolated cue-internal features like length and position do affect the learning of agreement patterns and have varied impact on the learning and generalization of agreement when factors such as prior word knowledge and noticing are taken into account. The implications for these findings are that specific knowledge of learned words may not be the same for the generalizations gleaned by learners which they apply to new words. This has serious repercussions for language learning. Specifically, testing learners' morphosyntactic knowledge of known words may provide inaccurate information regarding the generalizations they actually make for the entire morphosyntactic system of the language they are learning. Understanding the structural properties that may lead to differences in implicit and explicit learning and knowledge would lead to significant understanding in the current debates about the implicit and explicit memory and processing, as well as the possibility of transfer between these two knowledge types [54, 55].

Additionally, noticing lead to more accurate gender agreement for novel words. With this knowledge, one could see position and length of cues as mediating variables for noticing, in addition to having direct effects on unconscious/unaware learning. The importance of noticing, or at least the behavior of noticing, whether accurate or not, is important for learners to make broader generalizations of the target language's grammar. This links with the importance of awareness raising tasks in second language pedagogy [56, 57] and their relevance for learners' ability to extrapolate morphosyntactic patterns onto new words and structures [58, 59].

## Supporting information

**S1 Table. Determiner and noun item list.**
(DOCX)

**S1 File.**
(XLSX)

**S2 File.**
(XLSX)

**S3 File.**
(XLSX)

**S4 File.**
(XLSX)

## Author Contributions

**Conceptualization:** Daniel R. Walter.

**Data curation:** Daniel R. Walter, Galya Fischer, Janelle Cai.

**Formal analysis:** Daniel R. Walter, Janelle Cai.

**Investigation:** Daniel R. Walter, Galya Fischer, Janelle Cai.

**Methodology:** Daniel R. Walter, Galya Fischer, Janelle Cai.

**Project administration:** Daniel R. Walter, Galya Fischer.

**Resources:** Daniel R. Walter.

**Software:** Daniel R. Walter.

**Supervision:** Daniel R. Walter.

**Validation:** Daniel R. Walter, Galya Fischer.

**Visualization:** Daniel R. Walter.

**Writing – original draft:** Daniel R. Walter, Galya Fischer, Janelle Cai.

**Writing – review & editing:** Daniel R. Walter.

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
