## [Decision Letter · Decision Letter 0]

18 Jan 2024

PONE-D-23-39912The effect of cue length and position on noticing and learning of determiner-noun agreement pairings: Evidence from a cue-balanced artificial vocabulary learning taskPLOS ONE

Dear Dr. Walter,

Thank you for submitting your manuscript to PLOS ONE. After careful consideration, we feel that it has merit but does not fully meet PLOS ONE’s publication criteria as it currently stands. Therefore, we invite you to submit a revised version of the manuscript that addresses the points raised during the review process.

Thank you for submitting your valuable work.

The reviews, which are insightful and interesting, pointed to some unexplained aspects. The authors will notice the reviewers found merits in your study, but also raised several important concerns.

By my own reading, the manuscript still needs a lot of refinement, mostly related to soundness, conciseness and the control of confounding factors. Although this may sound counterintuitive, I am keen to understand authors' claims and keen on reading a refined manuscript.

We look forward to receiving your revised manuscript.

Kind regards,

Thiago P. Fernandes, PhD

Academic Editor

PLOS ONE

2. Your abstract cannot contain citations. Please only include citations in the body text of the manuscript, and ensure that they remain in ascending numerical order on first mention.

4. We note that you have referenced (ie. Hamrick, P. (2013)) which has currently not yet been accepted for publication. Please remove this from your References and amend this to state in the body of your manuscript: (ie “Hamrick, P. (2013)[Unpublished]”) as detailed online in our guide for authors

5. Please include captions for your Supporting Information files at the end of your manuscript, and update any in-text citations to match accordingly. Please see our Supporting Information guidelines for more information: http://journals.plos.org/plosone/s/supporting-infor

Reviewers' comments:

Reviewer's Responses to Questions

**Comments to the Author**

1. Is the manuscript technically sound, and do the data support the conclusions?

Reviewer #1: Yes

Reviewer #2: Partly

Reviewer #3: Partly

2. Has the statistical analysis been performed appropriately and rigorously? 

Reviewer #1: Yes

Reviewer #2: No

Reviewer #3: I Don't Know

3. Have the authors made all data underlying the findings in their manuscript fully available?

Reviewer #1: Yes

Reviewer #2: Yes

Reviewer #3: Yes

4. Is the manuscript presented in an intelligible fashion and written in standard English?

Reviewer #1: Yes

Reviewer #2: Yes

Reviewer #3: Yes

5. Review Comments to the Author

Reviewer #1: The paper presents a detailed study on the impact of cue properties in language learning.

The abstract provides a clear account of the paper, however the abstract's inclusion of specific references somewhat deviates from the ideal format of an abstract. While it is important to acknowledge foundational theories and prior research, detailed citations within an abstract can detract from its standalone comprehensibility.

Therefore, a more effective approach would be to summarize the key context and findings without direct references, ensuring that the abstract remains clear and self-sufficient for readers who may not have immediate access to the cited works or the full paper.

The Introduction and Literature Review sections are well-researched and provide a solid background. However, they might be too detailed, potentially overwhelming the reader with information. A more focused discussion on relevant studies directly linked to the research questions could improve readability.

The discussion on "Materials" (fourth paragraph onwards) (Page 12) seems to be repetitive considering the following section on "Procedure" (pages 13, 14, and 15).

The results of the study are presented with appropriate statistical analyses. However, there is an opportunity to make this section more accessible by simplifying the presentation of statistical data and by using visual aids.

In the Conclusion section, the practical implications of these findings could be highlighted more explicitly.

Additionally, please rectify the use of both first-person singular ("I") (Page 27) and first-person plural ("we") pronouns.

Also, I would like to alert the authors of some minor proof-reading errors that need to be corrected (example: post-test vs. posttest) and the use of serial comma.

Reviewer #2: Comments to the authors

Overall comments

The study presents an important investigation into the effects of cue length and position on learning and noticing in an artificial vocabulary task. While the data and commitment to open science are commendable, there are significant concerns. The lack of clarity in the data and R code on OSF, the choice of statistical models, and the insufficient explanation of how ‘noticing’ was identified and measured detract from the study’s impact. These major issues, alongside minor points like ambiguous terminology and potential oversights in the English agreement system, require careful revision. Addressing these points is crucial for enhancing the study’s clarity, analytical rigor, and contribution to the field.

Major issues

The data and R code on OSF

I am greatly appreciative that the authors follow the philosophy of open science and make all the materials available on OSF. I was particularly interested in the data and the R code used in the main analysis. I checked the OSF and there were various Excel files and two .R files that seem to be used for the data analysis. I just assumed that the files under "Revised data and R code for resubmit" are the ones reported in the current manuscript. I opened the "Cue Position Length data.xlsx" file. However, there are multiple sheets in it and I had to guess which sheet was used for the analysis. (It was actually easy for the first data set because the R code says "CueData <- as.data.frame(Cue_Position_Length_data)" but the second set is "novel test data"?).

In my opinion, there should be a CSV file that is loaded into R using "read.csv()" or "read_csv()". In this way, researchers who would like to replicate the analysis can easily identify which file should be used. It would also be beneficial for the community if the authors explain what each column represents and what they do in R, as well as demonstrating the necessary packages not only in the manuscript but also with the appropriate R code. The R code does not include any comments. I strongly recommend the authors put all the code into an R Markdown file and add plain text that explains the variables and steps the authors took. In short, just uploading the code and data is not enough to "reproduce all the analyses" (p. 16). For example, I found that "model 2b" reported in the manuscript does not appear in the "CueData.R" file, along with some other inconsistencies between the R code and the reported results. The authors should therefore review the analysis again and resolve any inconsistencies to improve the reliability of the data analysis.

GLMM vs. GLM

I am curious about the authors' choice to use a generalized linear model (GLM) instead of a generalized linear mixed-effect model (GLMM), which includes random effects. In my reanalysis of the data with GLMM using the “glmer()” function in the lme4 package, I found that GLMM provided a better fit than GLM. For more details on this analysis, please refer to the “replication-of-the-analysis.html” document. This additional perspective could potentially offer more insights and strengthen the overall analysis.

The identification of noticing

On page 17, where the authors discuss measuring participants' "noticing" through debrief questions, their methodology lacks a detailed explanation. I recommend that the authors provide a more thorough account of how they identified and differentiated levels of noticing. This explanation is crucial for understanding the measurement process and ensuring the validity of the findings related to participants' awareness and learning during the study.

Minor issues

On page 17 “models for main and mixed effects of cue Position and Length, in addition to Translation

Accuracy and Noticing scores.”

In my observation, the equations and R codes provided do not align with mixed-effects models as typically understood in statistical analysis. "Mixed effects" generally refers to models that include both fixed and random effects, capturing both population-level and individual variation. If the authors use the term "mixed effects" to mean "interactions" in a generalized linear model (GLM), this would be a deviation from the standard usage. It is important for the authors to clarify what they mean by "mixed effects" in this context to ensure the accuracy and clarity of their statistical approach.

On page 23 “The application of these findings is also important to discuss here.”

The term "[T]he application" is somewhat ambiguous. Based on my understanding, the authors' intention here seems to be "pedagogical implication." Therefore, I suggest using "pedagogical implication" instead of "the application." This change will help readers quickly grasp what the paragraph is about, ensuring clarity and enhancing the manuscript's readability.

On page 24 “the only agreement pattern that English speakers do use between determiners and

nouns is deciding between a/an,”

"[T]he only agreement pattern that English speakers do use between determiners and nouns is deciding between a/an" is a bit ambiguous. It could be interpreted to mean that "a/an" is the only form of agreement used in English, overlooking other forms of agreement like number agreement (as with "this/these" and "that/those") and usage based on countability (as with "some" for countable and uncountable nouns).

On page 24 “attempt overcome”

attempt to overcome?

Reviewer #3: The current study proposes that investigate the role of cue-internal factors – specifically position and length of cues - on noticing cues and learning determiner-noun agreement pairing. Though overall interesting, I believe the theoretical basis for this study is not clearly described and the study’s results could be better situated within the existing literature. I also find the lack of contr

---

## [Author Response · Author response to Decision Letter 0]

12 Mar 2024

So done

2. Your abstract cannot contain citations. Please only include citations in the body text of the manuscript, and ensure that they remain in ascending numerical order on first mention.

Citations removed from abstract

Added on page 5/6 under Methodology

4. We note that you have referenced (ie. Hamrick, P. (2013)) which has currently not yet been accepted for publication. Please remove this from your References and amend this to state in the body of your manuscript: (ie “Hamrick, P. (2013)[Unpublished]”) as detailed online in our guide for authors

Corrected

5. Please include captions for your Supporting Information files at the end of your manuscript, and update any in-text citations to match accordingly. Please see our Supporting Information guidelines for more information: http://journals.plos.org/plosone/s/supporting-infor

Caption added for supporting information

Reviewer #1: The paper presents a detailed study on the impact of cue properties in language learning.

The abstract provides a clear account of the paper, however the abstract's inclusion of specific references somewhat deviates from the ideal format of an abstract. While it is important to acknowledge foundational theories and prior research, detailed citations within an abstract can detract from its standalone comprehensibility. Therefore, a more effective approach would be to summarize the key context and findings without direct references, ensuring that the abstract remains clear and self-sufficient for readers who may not have immediate access to the cited works or the full paper.

condensed/citations removed

The Introduction and Literature Review sections are well-researched and provide a solid background. However, they might be too detailed, potentially overwhelming the reader with information. A more focused discussion on relevant studies directly linked to the research questions could improve readability.

The Introduction and literature review have been updated following many suggestions by other reviewers.

The discussion on "Materials" (fourth paragraph onwards) (Page 12) seems to be repetitive considering the following section on "Procedure" (pages 13, 14, and 15).

I decided to leave as is, with some slight edits to concisions, because the Materials section focuses on the make-up of the training and testing materials and the Procedure is a walk-through of how participants were presented with these materials. I think it is important to be detailed here for replicability. 

The results of the study are presented with appropriate statistical analyses. However, there is an opportunity to make this section more accessible by simplifying the presentation of statistical data and by using visual aids.

The statistical data was redone, as suggested by another reviewer to be a mixed-effects model. The tables have been updated and a visualization of predicted marginal values of agreement accuracy as an interaction between position and length has been added.

In the Conclusion section, the practical implications of these findings could be highlighted more explicitly.

The practical implications have been expanded.

Additionally, please rectify the use of both first-person singular ("I") (Page 27) and first-person plural ("we") pronouns.

Rectified to “we”

Also, I would like to alert the authors of some minor proof-reading errors that need to be corrected (example: post-test vs. posttest) and the use of serial comma.

Proof-reading errors corrected

Reviewer #2: Comments to the authors

Overall comments

The study presents an important investigation into the effects of cue length and position on learning and noticing in an artificial vocabulary task. While the data and commitment to open science are commendable, there are significant concerns. The lack of clarity in the data and R code on OSF, the choice of statistical models, and the insufficient explanation of how ‘noticing’ was identified and measured detract from the study’s impact. These major issues, alongside minor points like ambiguous terminology and potential oversights in the English agreement system, require careful revision. Addressing these points is crucial for enhancing the study’s clarity, analytical rigor, and contribution to the field.

These have been addressed (see below for specifics)

Major issues

The data and R code on OSF

I am greatly appreciative that the authors follow the philosophy of open science and make all the materials available on OSF. I was particularly interested in the data and the R code used in the main analysis. I checked the OSF and there were various Excel files and two .R files that seem to be used for the data analysis. I just assumed that the files under "Revised data and R code for resubmit" are the ones reported in the current manuscript. I opened the "Cue Position Length data.xlsx" file. However, there are multiple sheets in it and I had to guess which sheet was used for the analysis. (It was actually easy for the first data set because the R code says "CueData <- as.data.frame(Cue_Position_Length_data)" but the second set is "novel test data"?).

Data and R file in OSF has been updated and instructions included in the manuscript.

In my opinion, there should be a CSV file that is loaded into R using "read.csv()" or "read_csv()". In this way, researchers who would like to replicate the analysis can easily identify which file should be used. It would also be beneficial for the community if the authors explain what each column represents and what they do in R, as well as demonstrating the necessary packages not only in the manuscript but also with the appropriate R code. The R code does not include any comments. I strongly recommend the authors put all the code into an R Markdown file and add plain text that explains the variables and steps the authors took. In short, just uploading the code and data is not enough to "reproduce all the analyses" (p. 16). For example, I found that "model 2b" reported in the manuscript does not appear in the "CueData.R" file, along with some other inconsistencies between the R code and the reported results. The authors should therefore review the analysis again and resolve any inconsistencies to improve the reliability of the data analysis.

The code has been updated accordingly to include markdown and a read.csv input to ease replication.

GLMM vs. GLM

I am curious about the authors' choice to use a generalized linear model (GLM) instead of a generalized linear mixed-effect model (GLMM), which includes random effects. In my reanalysis of the data with GLMM using the “glmer()” function in the lme4 package, I found that GLMM provided a better fit than GLM. For more details on this analysis, please refer to the “replication-of-the-analysis.html” document. This additional perspective could potentially offer more insights and strengthen the overall analysis.

Thank you very much for this. I redid the entire statistical analysis with participant as a random effect, keeping the other variables as fixed effects. The findings regarding the best models remain the same but the variance was decreased and significance of the findings increased as a result of controlling for the participant variability.

The identification of noticing

On page 17, where the authors discuss measuring participants' "noticing" through debrief questions, their methodology lacks a detailed explanation. I recommend that the authors provide a more thorough account of how they identified and differentiated levels of noticing. This explanation is crucial for understanding the measurement process and ensuring the validity of the findings related to participants' awareness and learning during the study.

A paragraph was added under the Analysis section to describe the assignment of participants to the various noticing groups in more depth.

Minor issues

On page 17 “models for main and mixed effects of cue Position and Length, in addition to Translation

Accuracy and Noticing scores.”

In my observation, the equations and R codes provided do not align with mixed-effects models as typically understood in statistical analysis. "Mixed effects" generally refers to models that include both fixed and random effects, capturing both population-level and individual variation. If the authors use the term "mixed effects" to mean "interactions" in a generalized linear model (GLM), this would be a deviation from the standard usage. It is important for the authors to clarify what they mean by "mixed effects" in this context to ensure the accuracy and clarity of their statistical approach.

Apologies for the misuse of nomenclature. In the original paper, I was referring to interactions. Thank you for point that out. I have clarified this and made sure to be consistent with the redone stats that do include a mixed-model approach.

On page 23 “The application of these findings is also important to discuss here.”

The term "[T]he application" is somewhat ambiguous. Based on my understanding, the authors' intention here seems to be "pedagogical implication." Therefore, I suggest using "pedagogical implication" instead of "the application." This change will help readers quickly grasp what the paragraph is about, ensuring clarity and enhancing the manuscript's readability.

So changed

On page 24 “the only agreement pattern that English speakers do use between determiners and

nouns is deciding between a/an,”

"[T]he only agreement pattern that English speakers do use between determiners and nouns is deciding between a/an" is a bit ambiguous. It could be interpreted to mean that "a/an" is the only form of agreement used in English, overlooking other forms of agreement like number agreement (as with "this/these" and "that/those") and usage based on countability (as with "some" for countable and uncountable nouns).

Reworded to include number and count/non-count agreement

On page 24 “attempt overcome”

attempt to overcome?

corrected

Reviewer #3: The current study proposes that investigate the role of cue-internal factors – specifically position and length of cues - on noticing cues and learning determiner-noun agreement pairing. Though overall interesting, I believe the theoretical basis for this study is not clearly described and the study’s results could be better situated within the existing literature. I also find the lack of control regarding learner’s L2(s) concerning. Below are some suggestions for what I believe would improve this study.

These have been addressed (see below)

Major comments:

1. The initial introduction is quite clear and gives a good idea of why this is an adequate line of study and why it’s necessary to use an artificial language for such a study.

Small changes made to intro

2. The same cannot be said for the Literature Review, which I believe needs considerate restructuring. Although it seems that the content is pertinent, the reader is taken back and forth between paradigms and results in a list-like manner. While sometimes the authors synthesize and put findings into perspective, others the reader is “left hanging” a bit. I had to read this section three times to grasp the literature behind this study. I believe that if the authors pave a clearer path, readers who are less familiar with this literature will be able to follow more easily, giving them a better understanding of the significance of such a study. For instance, the paragraph between p. 8 and 9 seems to interrupt the flow of the description of Brooks et al.’s studies for no particular reason. All in all, it is hard to follow the logic behind this section and it would likely benefit from being separated into different sub-sections dedicated to different aspects of what is known regarding the contribution of cues to language comprehension and learning.

I decided not to separate into subsections but worked to make the relevant studies flow more logically from one another. I also deleted the study that “interrupted” the Brooks series of studies, as I felt it was unnecessary after the other revisions.

3. Given the importance of the learners’ language abilities for such a study, I find the following concerning:

- In the methods, it is stated that all of the learners had English as al L1, yet in the discussion, the authors state that “Most participants indicated that their first language was English”

- It seems that no note was take of learner’s knowledge of other languages, is this the case?

This data was collected and has been added to the participants section of the paper.

- Second languages could have a major impact on how learners process and learn linguistic cues – this has not been addressed.

The addition of the participant as a random variable should control for some of the variability from this. However, second language (or non-English L1) was not used as a control for in this study. I believe a more systematic comparison of specific L1s or L2s would be necessary rather than the random additional languages beyond English of these participants. This has been added to the limitations and future directions. 

4. Results. As I am only minimally familiar with Bayesian linear regression and cannot fully interpret the results.

- It would greatly facilitate reading and understanding if some sort of figure(s) of the results were provided.

Figures were added (see comment above to reviewer 2 for more details)

5. Discussion. I would have preferred to begin with a recap of the main idea behind the study before getting to the research questions. In line with my suggestions concerning the introduction, the discussion would benefit from a more systematic analysis of how the study’s findings fit and change the existing literature.

The discussion was updated to include these recommendations

Minor 

---

## [Decision Letter · Decision Letter 1]

3 Apr 2024

The effect of cue length and position on noticing and learning of determiner agreement pairings: Evidence from a cue-balanced artificial vocabulary learning task

PONE-D-23-39912R1

Dear Dr. Walter,

We’re pleased to inform you that your manuscript has been judged scientifically suitable for publication and will be formally accepted for publication once it meets all outstanding technical requirements.

Kind regards,

Thiago P. Fernandes, PhD

Academic Editor

PLOS ONE

Additional Editor Comments (optional):

Thank you for your careful edits.

Wishing you success with the study

Reviewers' comments:

Reviewer's Responses to Questions

**Comments to the Author**

1. If the authors have adequately addressed your comments raised in a previous round of review and you feel that this manuscript is now acceptable for publication, you may indicate that here to bypass the “Comments to the Author” section, enter your conflict of interest statement in the “Confidential to Editor” section, and submit your "Accept" recommendation.

Reviewer #1: All comments have been addressed

Reviewer #2: All comments have been addressed

Reviewer #3: All comments have been addressed

2. Is the manuscript technically sound, and do the data support the conclusions?

Reviewer #1: Yes

Reviewer #2: Yes

Reviewer #3: Yes

3. Has the statistical analysis been performed appropriately and rigorously? 

Reviewer #1: Yes

Reviewer #2: Yes

Reviewer #3: Yes

4. Have the authors made all data underlying the findings in their manuscript fully available?

Reviewer #1: Yes

Reviewer #2: Yes

Reviewer #3: Yes

5. Is the manuscript presented in an intelligible fashion and written in standard English?

Reviewer #1: Yes

Reviewer #2: Yes

Reviewer #3: Yes

6. Review Comments to the Author

Reviewer #1: I have read the revised version of the manuscript. The manuscript revision has addressed the points that were raised during the revision process.

Reviewer #2: I would like to express my sincere appreciation for the thorough and considerate response to the comments and suggestions made during this round of review.

Particularly, I am impressed with the efforts the authors have made to rerun the analysis and refine the R code as recommended. It is evident that a significant amount of work and dedication went into addressing the concerns raised, which has undoubtedly enhanced the quality and robustness of the study. The authors' commitment to rigorous scientific inquiry and openness to constructive feedback is commendable.

The modifications and additions the authors have made not only address the initial concerns but also significantly contribute to the clarity and depth of the research presented. It is clear that the revisions have strengthened the manuscript, making it a valuable contribution to the field.

Once again, I appreciate the authors' dedication to improving the manuscript based on the feedback provided.

Reviewer #3: I only have a few very minor comments:

p.5 often “affects”

p. 12 Instead of “a second langauge that was not English” it might be clearer to write “ a third language”

p.32 & 33 – I suppose this is a stylistic choice, but it seems strange to switch into present tense for the Results section.

p. 35 – I think the first paragraph should start with “The”.

7. PLOS authors have the option to publish the peer review history of their article (what does this mean?). If published, this will include your full peer review and any attached files.

Reviewer #1: **Yes: **Dr. Princy Pappachan

Reviewer #2: **Yes: **Yu Tamura

Reviewer #3: No

---

## [Editor Report · Acceptance letter]

8 Apr 2024

PONE-D-23-39912R1 

PLOS ONE

Dear Dr. Walter, 

I'm pleased to inform you that your manuscript has been deemed suitable for publication in PLOS ONE. Congratulations! Your manuscript is now being handed over to our production team.

Kind regards, 

on behalf of

Dr. Thiago P. Fernandes 

Academic Editor

PLOS ONE